# NeoRL: A Near Real-World Benchmark for Offline Reinforcement Learning

**Rong-Jun Qin**[1,2,*,*], **Xingyuan Zhang**[2,*], **Songyi Gao**[2,*],
**Xiong-Hui Chen**[1,2], **Zewen Li**[2], **Weinan Zhang**[3], **Yang Yu**[1,2,◇]
[1]National Key Laboratory for Novel Software Technology, Nanjing University
[2]Polixir Technologies
[3]Shanghai Jiao Tong University

## Abstract

Offline reinforcement learning (RL) aims at learning effective policies from historical data without extra environment interactions. During our experience of applying offline RL, we noticed that previous offline RL benchmarks commonly involve significant *reality gaps*, which we have identified include rich and overly exploratory datasets, degraded baseline, and missing policy validation. In many real-world situations, to ensure system safety, running an overly exploratory policy to collect various data is prohibited, thus only a narrow data distribution is available. The resulting policy is regarded as *effective* if it is better than the working behavior policy; the policy model can be deployed only if it has been well validated, rather than accomplished the training. In this paper, we present a **N**ear r**e**al-world **o**ffline **RL** benchmark, named NeoRL, to reflect these properties. NeoRL datasets are collected with a more conservative strategy. Moreover, NeoRL contains the offline training and offline validation pipeline before the online test, corresponding to real-world situations. We then evaluate recent state-of-the-art offline RL algorithms on NeoRL. The empirical results demonstrate that some offline RL algorithms are less competitive to the behavior cloning and the deterministic behavior policy, implying that they may be less effective in real-world tasks than in the previous benchmarks. We also disclose that current offline policy evaluation methods could hardly select the best policy. We hope this work will shed some light on future research and deploying RL in real-world systems.

## 1 Introduction

Reinforcement learning (RL) has shown impressive ability in simulated environments [1, 2]. However, current RL algorithms are hard to apply in real-world applications when there is no fast and cheap simulator. Such simulators provide virtual worlds for training RL agents by trial-and-errors which can incur huge losses in the real world. Instead, real-world systems often log data produced by an existing behavior policy, which we call the *working behavior policy* or *working policy*. A recent trend to alleviate the real-world trial-and-error costs is offline RL (also batch RL) [3], which aims to learn effective polices from pre-collected datasets. Thus, offline RL has the potential to refine the existing working policy with offline datasets produced by the working policy. To facilitate the development of offline RL approaches, benchmarks have been constructed involving datasets and protocols for evaluation [4, 5, 6]. RL Unplugged [4] uses datasets from the buffer of online training. D4RL [5] provides a diversity of datasets, including random, mixed medium datasets with replay buffer data or data collected by the expert policy, and datasets collected by humans and a planner. While it is a great contribution of these benchmarks to enable common playgrounds to test and compare

---

*These authors contribute equally. ◇ Correspondence to Yang Yu <yuy@polixir.ai>.

36th Conference on Neural Information Processing Systems (NeurIPS 2022) Track on Datasets and Benchmarks.

offline RL methods, our experience in applying RL in various real-world tasks, such as recommender systems, sales promotion, industrial control, etc., shows demands that are not satisfied by the previous benchmarks in several aspects as follows.

**Learning from conservative and limited data.** In many real-world domains, the dataset is conservative and limited. Take the urban power network system as an example, if the system can respond every 5 minutes and trajectories are split by weeks, the system only records about 52 trajectories per year [7]. More importantly, to ensure system safety and guarantee the utility, the working behavior policy that produced the data is conservative and near deterministic, since exploratory actions will harm the system or decrease the utility. Consequently, the offline data has a very narrow distribution. For instance, the recommender systems hardly try to present irrelevant items, but present the most relevant top-$n$ items to the user [8]. Note that conservativeness refers to less explorative and committing to the working policy, but does not prescribe the performance or the form of the working policy. In real-world tasks, the working policies can have different performances and forms (human-designed rules, a deep learning model, etc.), but are mostly conservative. Regarding the benchmarks, since exploration is quite necessary for RL training, previous benchmarks contain large and exploratory datasets. For example, the replay buffer from an online training process is often included [4, 5], which contains many random actions for exploration. Therefore, the exploration challenge, which we believe is a fundamental challenge for offline RL, is weakened in these benchmarks. During the experiments, we do find with exploratory datasets from previous benchmarks, offline RL algorithms are overestimated. However, the datasets from real-world scenarios are conservative, thus it is more challenging for offline RL algorithms.

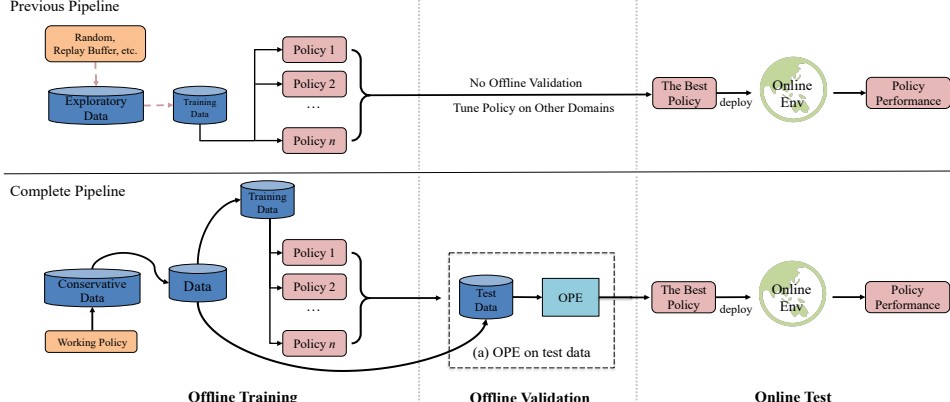

Figure 1: A complete pipeline of deploying offline RL should include phases of offline training, offline validation, and online test. In the offline validation, the offline policy evaluation is conducted on an extra test dataset. After validating, the best policy is chosen and deployed in the online environment. Previous benchmarks may use exploratory datasets and skip the offline validation step.

**Comparing with the working policy.** In real-world applications, an algorithm is thought to be *effective*, only when the trained policy is better than the existing working policy. Regarding the benchmarks, to facilitate offline learning, many of the datasets are collected from exploratory behavior policies or a mixture of multiple policies. In this case, behavior cloning (BC) as a baseline [4, 5] is often worse than the deterministic working policies. Consequently, the effectiveness of offline RL algorithms may be overestimated for their applicability if BC is used as a reference of the working policy on exploratory datasets.

**Offline policy validation is compulsory.** In almost all real-world tasks, we will train $n$ policies with different algorithms or hyper-parameters but are only allowed to choose one or two policies to deploy. The model can be deployed online *only* when we believe its online performance excels the existing working policy. Therefore, a process of "offline training → offline validation → online test" has been widely adopted, e.g., in recommender systems for supervised learning models [9]. Validation of RL policies using an offline dataset is also a challenge, due to the distribution shift incurred by the behavior and evaluation policies [6]. The recent work [10] proposes a practical offline workflow for conservative offline RL algorithms (e.g., CQL [11]) on robotic tasks, by utilizing comparative metrics across checkpoints and training runs. Regarding previous benchmarks, we notice that the

algorithm performances were all tested directly in the online environments once the training was accomplished [12, 11, 4, 5]. Offline validation has often been neglected, which bypasses the offline validation challenge. Consequently, even if an algorithm works well in the benchmark tasks, it is still hard to assess its performance in a real-world task. Offline policy evaluation (OPE) extends off-policy policy evaluation (OPPE) to offline setting [13, 14] and only uses the dataset to evaluate a policy, thus it can be applied as a tool for offline policy validation. It was found that running OPPE on the training dataset can cause biased estimations [15], so it will be more appropriate to conduct OPE on an unseen test dataset, which corresponds to the validation set in supervised learning.

To sum up, a complete pipeline of deploying offline RL should include offline training, offline validation, and online test phases, as shown in Figure 1.

**Our contributions and findings**. This paper proposes NeoRL, a suite of **N**ear r**e**al-world benchmarks for **o**ffline **RL**, to better reflect the above demands in real-world applications. NeoRL provides

1. **Conservative and limited datasets on all domains**. The datasets are collected from locomotion control, industrial control, financial trading, sales promotion and city management domains with real-world dataset properties.
2. **Online and offline selection benchmarks**. We benchmark recent model-free and model-based offline RL methods in NeoRL as a reference. We follow the complete pipeline for each training algorithm and provide the online selection results for comparisons.
3. **Comparison with the working policy**. To investigate whether these offline RL algorithms are effective, we compare them with the working policy. The experimental results show that they appear less effective when compared with the working policy.
4. **Empirical findings of offline evaluation**. Although offline policy evaluation before deployment is crucial, current OPE methods hardly help to select good policies in the offline RL setting.

By NeoRL, we not only propose new benchmarks, but more importantly, try to correct the way how offline data should be collected for offline RL benchmarks, and highlight the complete pipeline for deploying offline RL. The datasets from real-world systems are collected by a more conservative policy. The policies trained by offline RL methods should be validated and selected offline rather than directly deployed in the system when the training stops. The offline training and offline policy evaluation modules have been integrated into the NeoRL code repository to help run the full pipeline easily. Codes for the data-collection process and benchmark results can be found at `https://github.com/polixir/NeoRL`.

## 2   Offline Reinforcement Learning

Online RL algorithms interact with the environment and update the policy to get a high episodic return. In the offline RL setting, the environment is not provided during training, and only a batch of data is accessible. For simplicity, the data can be viewed as collected by a single behavior policy $\pi_b$.

Although off-policy algorithms can be readily applied to a replay buffer, running an off-policy RL algorithm on a static buffer can sometimes diverge, due to issues like the distribution shift [16]. To learn a robust policy, recent offline RL algorithms explicitly or implicitly prevent the training policy from being too disjoint with $\pi_b$ [12, 11, 17, 18]. Besides, the absence of a cheap environment also makes it untamed to evaluate a training policy and select a good one. Unlike traditional RL policy validation that directly runs the policy in the environment to evaluate the ground-truth value and then selects the appropriate hyper-parameters and model, in general, tuning offline RL algorithms requires offline policy evaluation (OPE). Most off-policy policy evaluation (OPPE) [19] methods can be used as OPE tools with little change. However, OPPE methods are primarily designed to approximate the value of a policy, while in the offline RL setting, we expect an offline policy evaluation (OPE) method can select good policies during the training. Therefore, an OPE method will be effective if it tells the correct ranks of the candidate policies, rather than approximating the values.

## 3   Previous Benchmarks and Recent Offline RL algorithms

Recently, some offline RL benchmarks have been proposed to facilitate the research and evaluation of offline RL algorithms. These benchmarks include multiple aspects of offline tasks and datasets, and also the performances of prior offline algorithms on these tasks [4, 5, 20]. The celebrated Atari

57 games and Gym-MuJoCo tasks (or DeepMind Control Suite [21]) have been widely used to benchmark online and offline RL methods. Besides these two domains, D4RL [5] also releases offline datasets of mazes, FrankaKitchen [22], and offline CARLA [23]. These datasets are designed to cover a range of challenging task properties in real-world scenarios, including narrow and biased data distributions, multi-task data, sparse rewards, etc. RL Unplugged [4] includes datasets from Atari and DeepMind control suite, where the properties of these tasks range from different action and observation spaces, partial observability, etc. Other aspects of real-world challenges are also noted in [24]. Although the properties of tasks are well covered in D4RL and RL Unplugged, the properties of the real-world datasets are underexplored. To guarantee the system stability and performance, datasets from real-world systems cannot be too exploratory. Therefore, the out-of-distribution problem [25] is more challenging for offline RL when faced with real-world datasets. Recent offline algorithms utilize the training buffer data to run offline RL algorithms [12, 26, 16], and D4RL and RL Unplugged contain data from the training buffer. Intuitively, a wider data distribution weakens the exploration challenge, thus the offline RL algorithms may be overestimated on the datasets that are unlike to meet in the real world. Meanwhile, offline RL methods are often pessimistic about the out-of-data distribution, by constraining the RL agent to be close to the offline data [12, 17], or only trusting the learned environment model when the uncertainty of the generated data is low [27, 28]. In D4RL and RL Unplugged, it is noticed that online policy selection is not allowed so that they propose evaluation protocols that just train with the optimal hyper-parameters from the similar domains and do not tune hyper-parameters anymore. This protocol implicitly requires a transferable environment which contradicts practical offline RL setting, since we can learn from these domains and adapt to the online environment as done in [29, 30]. Besides, these policies are directly evaluated in the environment once finishing the training. Comparisons with previous benchmarks are listed in Table 5 in B.

DOPE [6] is to benchmark the performance of off-policy policy evaluation methods and tested on RL Unplugged and D4RL. We do find SOTA OPPE methods from the DOPE benchmark, i.e., FQE and IS will fail when all the trained candidate policies have similarly poor performance. Additional discussions and demonstrations can be found in B.

## 4   Near Real-World Benchmarks

We only assume that the working policies are sub-optimal and conservative, which are often common in realistic applications but are not well embodied in previous benchmarks. Therefore, we produce policies to have these two properties. Most importantly, we follow the full training and validation pipeline, conducting OPE for policy selection on the different test datasets. In real-world scenarios, the dimension of the state and action space can be high and the transition functions are complex, with stronger stochasticity. Hence, besides the widely-used locomotion controlling tasks, we include tasks that are high dimensional or with high stochasticity from some real-world scenarios, i.e., industrial controlling, financial trading, etc.

### 4.1   Collect Datasets with Conservative Policies

The historical interaction data collected from the real world are often produced by (deterministic) working policies, rather than a random policy or replay buffers. Since the actions are continuous in all domains, we use the Gaussian distribution as the policy output. To simulate the real-world data-collection scenarios, we adopt the following steps to collect the data:

1. **Obtain data-collection policies**. For each environment, we use SAC [31] to train on each environment until convergence and record a policy at every epoch. We denote the policy with the highest episodic return during the whole training as the expert policy. Another three levels of policies with around $25\%, 50\%, 75\%$ expert performance are stored to simulate multi-level sub-optimal policies, denoted by low, medium, and high respectively.
2. **Collect data**. With probability $20\%$, we sample from the trained Gaussian policies to execute, otherwise, we use the mean of Gaussian to execute. We sample from the Gaussian policy during the data collection for two reasons: (1) Because of human manipulation errors, the action demonstrations may be noisy, so we use samples from the policy to reproduce this phenomenon. (2) For training offline RL algorithms: When the transition function is deterministic (e.g., Gym-MuJoCo), the deterministic policy may produce similar and even repetitive trajectories. However,

if the transition function is stochastic, we only execute a deterministic policy (mean of the trained Gaussian policy) to generate the datasets.

3. **Make training and test datasets**. For each level, 4 policies with similar returns are selected, among which three policies are randomly selected to collect the training data used for offline RL policy training, and the left one produces the test data for offline validation. The default size of the test data is $1/10$ of the training data for each task. The extra test dataset can be used to design the offline evaluation method for the model selection during training and hyper-parameter selection.

It should be noted D4RL collects the data by sampling from the policy output distribution each step or even mixes data from the training buffer, which collects more exploratory data. We construct datasets of 100 trajectories as the limited data setting. To help verify the impact of different amounts of data, for each task, we provide training datasets of three-level sizes of $10^2$, $10^3$, and $10^4$ trajectories by default. See A for detailed sample sizes.

## 4.2 Benchmarks with Online and Offline Policy Selection

We benchmark some recent offline RL algorithms on the proposed datasets under the full pipeline of deploying offline RL, and provide online selection results as a reference. The online selection is contained because the performance via online selection can reflect the upper bound of an algorithm, and would help once OPE or other approaches can select the optimal policy without interacting with the environment. We then follow the fully offline training pipeline and benchmark these algorithms, where the policy model is selected by offline policy evaluation (OPE) methods on the test dataset. Especially, to verify whether offline RL methods are effective (exceed the working policy), the comparisons with the deterministic version of $\pi_b$ are involved.

## 5 Tasks and Datasets

As explained in D4RL, we also use high-quality simulators for data collection and evaluation.

**The industrial benchmark (IB)** [32] is an RL benchmark environment motivated to simulate the characteristics presented in various industrial control tasks, such as wind or gas turbines, chemical reactors, etc. It includes problems commonly encountered in real-world industrial environments, such as high-dimensional continuous state spaces, delayed rewards, complex noise patterns, and high stochasticity of multiple reactive targets. Since the IB environment is high-dimensional and highly stochastic, we use the mean of Gaussian policy when collecting data, rather than sampling from it.

**FinRL** environment [33] provides a way to build a trading simulator that replicates the real stock market and supports backtesting with important market frictions such as transaction costs, market liquidity, investor risk aversion, and so on. In FinRL, per trading day can trade once for the 30 stocks in the pool. The reward function is the difference in the total asset value between the end of the day and the previous day. The environment may evolve itself as time elapses. Because the training dataset of $10^4$ trajectories is too large, we only provide $10^2$ and $10^3$ trajectories for FinRL.

**CityLearn (CL)** environment [34] reshapes the aggregation curve of electricity demand by controlling energy storage in different types of buildings. The objective is to coordinate the control of domestic hot water and chilled water storage by the electricity consumers (i.e., buildings) to reshape the overall curve of electricity demand. This environment is highly stochastic and with high-dimensional space.

**SalesPromotion (SP)** environment simulates a real-world sales promotion platform, where the platform operator delivers different discount coupons to each user to promote the sales each day. The number of coupons and the discount the user received will affect his behavior. A higher discount will promote the sales, but the cost will also increase. The goal for the platform operator is to maximize the total income. This environment is partly built on our (Polixir) real-world sales promotion project. We only provide historical data made by a human operator and real users on the sales promotion platform for this domain.

**Gym-MuJoCo** [35] environments are the standard testbeds for online RL algorithms. We select three domains and construct the offline RL tasks, i.e., HalfCheetah-v3, Walker2d-v3, and Hopper-v3. The 3 selected tasks are widely used in existing benchmarks, so we introduce the conservative and limited data properties into these tasks to investigate the impact on previous benchmarking results.

For each domain except FinRL and SP, NeoRL contains 9 tasks (3 levels of behavior policy performances and 3 kinds of sizes). Thus, NeoRL currently contains 7 domains with 52 tasks in total. Detailed features of these environments can be found in the A.

# 6 Experiments

In the experiments, we focus on the following questions (Q): ***Q1***: If we have perfect offline policy selection, are current offline RL algorithms *effective* (better than the deterministic working policy) on NeoRL datasets? ***Q2***: What is the influence of the conservative datasets? ***Q3***: What is the influence of limited sizes of datasets? ***Q4***: If we just use current off-policy evaluation methods to validate the trained policies before deployment, can we get the *effective* policy?

To answer ***Q1***, we choose the best online performance on each task for each algorithm among the given hyper-parameters, and compare them with the baselines. We compare results on Gym-MuJoCo domains of NeoRL and D4RL to answer ***Q2***. We investigate the influence of different sizes of datasets to answer ***Q3***. For ***Q4***, we follow the complete pipeline of deploying offline RL, conducting the offline policy selection and validation via OPE methods.

We re-implement several algorithms to easily call them by a unified interface. The re-implementation has been verified on Gym-MuJoCo-medium tasks from D4RL dataset and matches the original results (see Table 6 in D). We roughly divide these algorithms into two categories: model-based and model-free. Since offline RL algorithms are sensitive to the choice of hyper-parameters, we conduct a grid search on hyper-parameter space to choose the best policy. Details of the hyper-parameters settings are in F.

## 6.1 Comparing Methods

We consider four non-learning baselines and then learning methods.

**Expert** We run SAC until convergence in each environment to choose the policy with the highest returns and call it *expert*. Expert is used as a reference for a good policy. However, it does not imply that the expert is optimal. While for the sales promtion domain, we use a human operator policy.

**Random policy** The uniform distribution over the action space is a valid but very weak policy on almost all tasks. It is usually used in normalizing the policy performance.

**Deterministic Policy** Commonly, the running system involves a deterministic working policy. We take the deterministic behavior policy as the working policy in our experiments. An offline trained policy is thought *effective* only if its performance is better than this working policy.

**Behavior Policy** The behavior policy is used to collect the data. If the offline data-collection process has no randomness injected, the behavior policy equals the deterministic policy.

**Compared Learning Methods** Behavioral cloning (BC) trains a policy to imitate the behavior policy from the data. We treat BC as a baseline of learning methods. We also includes model-free methods BCQ [12], PLAS [18], CQL [11] and CRR [17], and model-based methods BREMEN [36] and MOPO [27]. An overview of these compared algorithms can be found in the Appendix C.

## 6.2 Evaluation Methods

We only keep the policy at the last epoch for each hyper-parameter configuration and seed, except for BC where mean squared error can be used as the early-stop criterion (see F for details).

**Online policy selection evaluation.** Online selection results can be treated as the upper bound of the algorithm performance, despite it favors algorithms with more hyper-parameters (noted in [4]). Each trained policy is evaluated for 1,000 episodes to approximate the ground-truth performance. The final performance is reported for the best hyper-parameter with the highest average score over 3 seeds. Although the result with more seeds is appealing, empirical results show the average performance of the compared algorithms did not change too much. However, running with one more seed takes over 7,000 GPU hours (see E for details).

**Offline policy selection.** In general, we only have very few chances to deploy trained policies in real-world systems. Even though the trained policies only differ in random seeds, they will be treated

as different policies. Thus, we conduct OPE on the extra test dataset to select the best policy among policies trained by different hyper-parameters and seeds, then we report their online performance. An effective OPE method only needs to tell the relative performance between policies to select the best model. Specifically, we choose two representative OPE methods: fitted Q evaluation (FQE) [37] and weighted importance sampling (WIS) [38]. FQE takes a policy as input and performs policy evaluation on the fixed dataset by Bellman backup. After learning the Q function of the policy, the performance is measured by the mean Q values on the initial states from the dataset and actions by the policy. WIS is a canonical variant of important sampling (IS). IS only uses the ratio between target policy and behavioral policy to weight the episodic reward in the dataset, while WIS can further reduce the IS variance. Both methods are run with 3 seeds on the candidate policy set. The three non-learning baselines do not need OPE. Implementation details of OPE can be found in G.

## 6.3 Results

We calculate an average rank and average normalized scores respectively. The rank of an algorithm or baseline is determined by the score on each task, and the final average rank is computed over the 52 tasks. The average rank of each algorithm is shown in Table 1, and the average normalized scores are shown in Figure 2, for online and offline evaluation respectively. Detailed raw scores and normalized scores of each task are deferred to H. The normalization $100 \times \frac{\text{raw score} - \text{random score}}{\text{expert score} - \text{random score}}$ is also adopted in our evaluation. One point should be highlighted that the average (or any kind of aggregated) performance is only given as a convenient metric. Since the properties of each domain are quite different, they should be seen as a collection of individual tasks.

Table 1: Average ranks over 52 tasks of online, FQE, WIS policy selection results.

| Name | Det. policy | Behavior policy | Random | BC | BCQ | PLAS | CQL | CRR | BREMEN | MOPO |
|------|-------------|-----------------|--------|-----|------|------|------|------|--------|------|
| Online | 4.88 | 5.67 | 8.90 | 5.04 | 6.08 | 5.23 | **2.19** | 4.00 | 5.31 | 7.69 |
| FQE | 3.33 | 3.92 | 8.48 | **3.33** | 6.10 | 6.56 | 4.35 | 4.56 | 6.25 | 8.13 |
| WIS | 3.69 | 4.42 | 8.56 | **3.77** | 5.85 | 5.56 | 4.60 | 4.42 | 6.25 | 7.88 |

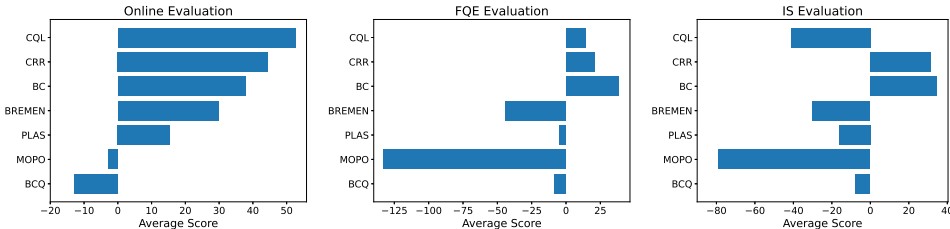

Figure 2: Average normalized score of each algorithm on 52 tasks by online selection and offline policy selection.

**Online policy selection results. (The answer to *Q1*)** From Table 1, CQL achieves the highest rank of 2.19, which greatly outperforms other algorithms. BC matches the performance of the deterministic policy, indicating BC recovered the deterministic behavior policy from the datasets. Intriguingly, the results demonstrate BC is a strong baseline: the other six offline RL algorithms fail to outperform BC in 152 out of 312 comparisons (note that we have set the quality of datasets to three levels where BC is believed to perform poorly in the low-quality dataset). Using the Nemenyi test [39], the critical difference of 10 comparing methods over 52 tasks with a confidence level 95% is 1.8787. Therefore, if we take BC as the reference, only CQL is significantly better than BC, while Random and MOPO are significantly worse. The result is the same if we take the deterministic policy as the reference, implying current offline RL algorithms are less effective. The winning rates against behavior policy, the deterministic policy, and BC for each compared baselines can be found in Table 23 in the Appendix H.

For model-based approaches, the overall online performance is worse than model-free methods, but they can bring remarkable improvements in some domains. For instance, on HalfCheetah-Low and HalfCheetah-Medium tasks, BREMEN and MOPO can outperform other algorithms and baselines

Table 2: The difference of the normalized scores between each algorithms and the behavior policy (dataset returns) on Gym-MuJoCo medium tasks.

| Task Name | BCQ | PLAS | CQL | MOPO |
|---|---|---|---|---|
| HalfCheetah-D4RL | 6.6 | 8.1 | 10.3 | 6.1 |
| HalfCheetah-NeoRL | 4.6 | 4.8 | 8.6 | 16.3 |
| Hopper-D4RL | 22.5 | 4.9 | 54.6 | −5.5 |
| Hopper-NeoRL | 5.7 | 19.2 | 22.5 | −41.0 |
| Walker2d-D4RL | 42.3 | 56.1 | 63.7 | 3.2 |
| Walker2d-NeoRL | 18.7 | −8.4 | 14.3 | −3.1 |

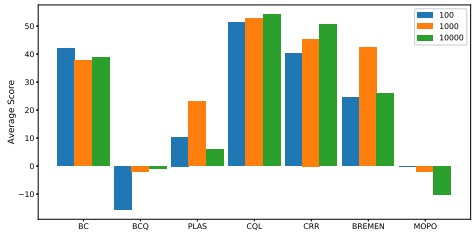

Figure 3: Average normalized score of each algorithms with respect to the number of trajectories.

by a large margin, which reveals the potential of model-based offline RL approaches. However, the dataset can be less diverse as the quality improves, which may incur bias [40] in environment learning and lead to poorer performance on high-quality datasets.

**The influence of conservativeness. (The answer to *Q2*)** We calculate the difference of normalized score between the online performance of compared algorithms and the behavior policy performance (dataset episodic returns) on the closest Gym-MuJoCo tasks and summarize the results in Table 2. The performance on D4RL is directly adopted from D4RL or the original paper. It can be observed from Table 2, 10 out of 12 results are overestimated when compared with the behavior policy. However, datasets from running systems are mostly conservative, thus we should evaluate offline algorithms on conservative datasets to avoid overstating the applicability of current offline RL algorithms.

**The influence of different sizes of datasets. (The answer to *Q3*)** We also evaluate the performance of each algorithm with respect to the number of trajectories used in training. As shown in Figure 3, for 5 out of 7 algorithms, the performance by online policy selection grows as the training number increases from 100 to 1000. However, only performances of BCQ, CQL, CRR increased when the training trajectories further increased to 10000. We note that the smaller size of datasets can decrease the performance, while providing more conservative data will not help too much. The rationale may be that the conservative dataset presents a rather narrow distribution, thus even trained with 10,000 trajectories, the performance will not improve too much. BREMEN has gained an overall significant improvement from 100 to 1000 episodes, implying that a large dataset helps build better models and benefits model-based offline RL algorithms. However, providing 10,000 trajectories decreases the performance of BREMEN. The same trend can be observed in MOPO. We notice that both BREMEN and MOPO perform poorly in the IB environment, where learning an environment model may be unprecedented challenging in such environments if we only use MSE as the metric.

**Offline policy selection results. (The answer to *Q4*)** However, the result of offline selection favors BC. From Table 1 and Figure 2, for both OPE methods, the average rank and average normalized score of BC become the best. That means if we follow a strict offline setting and the full offline pipeline, current offline RL algorithms are no better than the naive BC and the deterministic policy. Except CQL and CRR, other learning algorithms significantly fall behind BC (see Table 24 and 25 in H for winning rates). From the normalized scores over three evaluations, on over a half of tasks, online evaluation, and two OPE could not reach an agreement on the best algorithms and policies. We also plot the estimate vs ground-truth return as [14, 6] in the Appendix G. From Figure 8 to 14 in G, one rationale for the disagreement of online and offline evaluation is that when the candidate set contains many extremely low-performance policies, FQE and WIS cannot distinguish them well (FQE and WIS can give an either high or low evaluation to a policy with low online performance, thus the estimated returns distribute vertically). However, FQE and WIS may also fail when the ground-truth returns are very different (estimated returns distribute horizontally). A possible reason for the failure of the two OPE methods may be that they are originally designed to approximate the ground-truth performance, while it is challenging to estimate the ground-truth value when the state-action space coverage is small. However, we only need the OPE methods to tell the ranks of a set of policies to select the best policy. Empirically, we may benefit from OPE if it is possible to preclude those poor policies or make them more distinguishable. Detailed results of OPE can be found in G.

# 7 Closing Remarks

We benchmark recent SOTA offline RL algorithms on NeoRL tasks, including model-free and model-based algorithms. The experimental results demonstrate that these compared offline RL algorithms fail to outperform neither the naive BC nor the working policy on NeoRL, only except CQL with online policy selection. However, offline RL algorithms aim to refine the working policy. Thus, we present the following aspects that may facilitate the applicability of offline RL:

- **How could offline data be collected for benchmarks?** The real-world offline data are always conservative. Although for the purpose of training offline RL algorithms, we use slightly perturbed deterministic policies when collecting the data, other approaches may be adopted, e.g., only add noise to the observation while maintaining a fully deterministic policy. D4RL adopted human-designed planners to collect data or human expert demonstrations, which can also collect conservative data. However, it requires some domain knowledge. Besides, it is feasible to first learn an environment model and then use model predictive control (MPC) [41] to collect the data.
- **The future design of offline RL algorithms.** The experimental results further show that current model-based offline RL approaches are overall worse than model-free approaches. However, we find model-based algorithms may get a higher score when the performance of the behavior policy is low and the size of data is small. Besides, the learned environment model can be used to conduct policy evaluation. With the two empirical observations, model-based offline algorithms may have a better potential for real-world scenarios. Meanwhile, we have noticed that better model-learning approaches based on adversarial training [42, 43, 44] could help. Although pessimism makes the policy safe, we argue it also constrains the policy to be close to the working policy, thus offline RL algorithms should delicately use the pessimism if they aim to refine the current working policy.
- **How does OPE help validate the policy?** With a list of offline trained policies, the offline validation only needs to select the best policy in the list and tell whether the best policy exceeds the current working policy. Future OPE methods can extract pairwise or listwise features to predict the correct rank of the policies, rather than estimating the ground-truth returns.

In the future, we will investigate more real-world offline RL challenges, by constantly providing new near real-world datasets and tasks. We also hope the NeoRL benchmark will shed some light on future research and draw more attention to deploying real-world RL applications.

## Acknowledgments and Disclosure of Funding

This work was supported by National Key Research and Development Program of China (2020AAA0107200) and National Natural Science Foundation of China (61921006, 61876077, 62076161). We also thank Zhen Xu and Shengkai Huang for making early versions of the datasets.

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
