# OpenReview forum: "NeoRL: A Near Real-World Benchmark for Offline Reinforcement Learning"
_NeurIPS.cc/2022/Track/Datasets_and_Benchmarks — NeurIPS 2022 Datasets and Benchmarks _

### Official Review · Reviewer_qEGL · 2022-07-09

**Rating:** 10
**Confidence:** 5
**Correctness:** I cannot see any technical incorrectn…
**Clarity:** The paper is well written

**Strengths:**

1. NeoRL includes diverse datasets of real-world tasks, which are more conservative and limited than the previous ones.

2. NeoRL introduces offline validation in the offline training pipeline, which is not formally considered in the previous ones.

3. To clearly investigate the effectiveness of offline RL algorithms, NeoRL compares the performance of learned policy with working policy, rather than behavior clone policy.

4. Based on the proposed pipeline of offline training, the authors re-evaluate many existing offline algorithms.

**Weaknesses:**

I have a question about policy evaluation. In CV tasks, researchers will select the best model in the validation set. However, in reinforcement learning, the convergence is the most important, and researchers usually care about the performance of the converged policy. We know the best policy might not be the converged policy. So, in offline RL, should we focus on the best policy or the converged policy?

**Additional Feedback:**

We know offline RL is a fast-growing field, there are many existing algorithms. More baselines are expected to be evaluated.

**Documentation:**

The documentation is sufficient and clear.

**Relation To Prior Work:**

The relation and difference between this work and previous offline benchmarks, e.g., D4RL, are very clear.

**Summary And Contributions:**

In this paper, the authors propose a novel benchmark for offline reinforcement learning, NeoRL. NeoR includes real-world tasks, e.g., industrial benchmark, FinRL, CityLearn, SalesPromotion. The datasets in NeoRL are conservative and limited, which follows the properties of real-world datasets. The authors propose a new method to evaluate the performance of offline RL algorithms. Instead of comparing with the behavior clone, they propose to compare with the working policy. The benchmark introduces an offline validation phase, which evaluates the learned policy using a test dataset before deploying it to the environment. A lot of existing offline methods are evaluated in the benchmark. NeoRL is more practical and challenging compared with previous benchmarks, and I believe that it will become a popular benchmark for offline learning.

---

> ### Author Response · Authors · 2022-08-10
> **Response to reviewer qEGL**
>
> Q1: "We know the best policy might not be the converged policy. So, in offline RL, should we focus on the best policy or the converged policy?"
>
> A1: We believe "select the best model in the validation set" in CV tasks is the right way in practice. It is not only to select the model but also to estimate how the model will perform after the deployment. Unfortunately, this way is not followed, also not easy to follow, in offline RL. Obviously, we should select the best model. However, the current offline evaluation methods are incapable of evaluating these models accurately, since the policies mostly act out-of-the-data. This is the challenge that NeoRL benchmark tries to highlight.
>
>
>
> Q2: "We know offline RL is a fast-growing field, there are many existing algorithms. More baselines are expected to be evaluated."
>
> A2: Thanks for the insightful suggestions. We are also constantly updating the GitHub [repo](https://github.com/polixir/NeoRL) to include some new offline RL algorithms as well as offline evaluation methods.

---

### Official Review · Reviewer_UDpC · 2022-07-23
**A meaningful benchmark for deploying real-world RL applications.**

**Rating:** 7
**Confidence:** 4
**Correctness:** It seems correct to me.
**Clarity:** The paper is well written.

**Strengths:**

I think the topics of learning from conservative data and offline evaluation are very meaningful. They are important in deploying real-world RL applications.

Since many real-world policies cannot be too exploratory, I agree with the authors that a good offline policy should perform well on conservative data. The results show that many offline RL algorithms may be overestimated, which will inspire future works.

The paper is written clearly.

**Weaknesses:**

The authors propose a complete pipeline for deploying offline RL in Fig.1, but they don't follow such a pipeline to evaluate the RL algorithms. Instead, they show the results of online evaluation and offline evaluation respectively. And the results show that online evaluation and offline evaluation could not reach an agreement. It is confusing if the policy selected by offline evaluation is better than the deterministic working policy.


**Additional Feedback:**

1. I think the authors could try some more offline evaluation methods.

2. I think it is better to follow the pipeline proposed in Fig.1 to evaluate the offline RL algorithms, including offline selection, and evaluate the best policy online.

**Documentation:**

The documentation is detailed.

**Ethics:**

There are not any ethical concerns.

**Relation To Prior Work:**

The authors discuss the relation to prior works sufficiently.

**Summary And Contributions:**

The paper presents a near real-world offline RL benchmark (NeoRL). It concentrates on learning from conservative and limited data, because many real-world working policies are conservative. In addition, it evaluates the offline policies in both offline and online ways, where offline evaluation is more reasonable in a real-world application. The authors investigate popular offline RL algorithms in the near-real setting and find that many offline RL algorithms may be overestimated.

---

> ### Author Response · Authors · 2022-08-10
> **Response to reviewer UDpC**
>
> Q1: "The authors propose a complete pipeline for deploying offline RL in Fig.1, but they don't follow such a pipeline to evaluate the RL algorithms. Instead, they show the results of online evaluation and offline evaluation respectively. And the results show that online evaluation and offline evaluation could not reach an agreement. It is confusing if the policy selected by offline evaluation is better than the deterministic working policy."
>
> A1: We believe Fig.1 is the correct pipeline in practice and the offline evaluation results strictly follow this pipeline. In our experiments, we conduct online evaluations in order to show that the current offline evaluation methods are not quite effective. We would like to clarify that the results of the offline evaluation are indeed the results of offline policy selection (see Section 6.2 for evaluation methods). We have revised the paper to make this point clear. As a result, there is no doubt that the offline selected policy is worse than the deterministic working policy.
>
>
>
> Q2: "I think the authors could try some more offline evaluation methods."
>
> A2: Thanks for the insightful suggestions. We are also constantly updating the GitHub [repo](https://github.com/polixir/NeoRL) to include some new offline RL algorithms as well as offline evaluation methods.

---

### Official Review · Reviewer_9YLZ · 2022-07-25
**This article is well written and many experiments have been designed.**

**Rating:** 7
**Confidence:** 4
**Correctness:** All my concerns have been mentioned a…
**Clarity:** This article is well written and illu…

**Strengths:**

1. The authors collected a lot of real world data, including locomotion control, industrial control, financial trading, sales promotion and city management domains with real-world dataset properties.
2. To investigate whether these offline RL algorithms are effective, the author compare them with the working policy.
3. The author adds a section on offline policy validation, which feels inspired.

**Weaknesses:**

1. Why did you only choose these three environments? I see that the commonality of these three environments is relatively simple. HalfCheetah-v3, Walker2d-v3, and Hopper-v3. Why not choose Humanoid, whose state space 376 and action space 17.
2. I have great doubts about the author's point of view, and I hope the author can further elaborate on this point of view.
    “Comparison with the working policy.The experimental results show that they appear less effective when compared with the working policy.”



**Additional Feedback:**

I has presented it in the previous section.

**Documentation:**

Good. The instructions on how to set up the environment are given in the appendix. Also, github url is provided.

**Ethics:**

There are no any ethical concerns that warrant further discussion or review.

**Relation To Prior Work:**

The paper makes good connections to many prior works.

**Summary And Contributions:**

Previous offline RL benchmarks commonly involve significant reality gaps, which include rich and overly exploratory datasets, degraded baseline, and missing policy validation. This paper proposes NeoRL, a suite of Near real-world benchmarks for offline RL.

---

> ### Author Response · Authors · 2022-08-10
> **Response to reviewer 9YLZ**
>
> Q1: "Why did you only choose these three environments? I see that the commonality of these three environments is relatively simple. HalfCheetah-v3, Walker2d-v3, and Hopper-v3. Why not choose Humanoid, whose state space 376 and action space 17."
>
> A1: We use these 3 tasks in order to have a comparison with previous benchmarks such as D4RL. Meanwhile, our focus is not solely on the MuJoCo locomotion controlling tasks. Besides, we included some domains that have complex transition functions (IB), high observation space (IB, FinRL), and high action space (FinRL, CL). The observation and action space are pasted below for convenience (see also Table 3 in Appendix A):
>
>
>
> | Env Name       | Observation Space | Action Space | Max Time Steps |
> | -------------- | ----------------- | ------------ | -------------- |
> | HalfCheetah-v3 | 18                | 6            | 1000           |
> | Hopper-v3      | 12                | 3            | 1000           |
> | Walker2d-v3    | 18                | 6            | 1000           |
> | IB             | 180               | 3            | 1000           |
> | FinRL          | 181               | 30           | 2516           |
> | CL             | 74                | 14           | 1000           |
> | SP             | 4                 | 2            | 50             |
>
>
>
>
> Q2: "I have great doubts about the author's point of view, and I hope the author can further elaborate on this point of view. “Comparison with the working policy.The experimental results show that they appear less effective when compared with the working policy.”"
>
> A2: We noticed that in previous benchmarks, the datasets are collected with exploration-enabled policies. The scores of these baselines can usually be improved simply by turning off the exploration, which is exactly what will be done in real-world tasks. Therefore, we pointed out in the paper that algorithms should be compared with the working policy, i.e., the policy with the exploration turned off.

---

### Official Review · Reviewer_uqJ5 · 2022-07-27

**Rating:** 5
**Confidence:** 4
**Clarity:** Yes

**Strengths:**

1. It is valuable to consider more conservative data composition in offline RL, which matches the need of many real-world tasks.

2. The authors compare the performances of current methods under online evaluation and offline evaluation respectively. This is of great value to the community since performing offline evaluation is pivotal in offline RL research.

3. The tasks constructed in the paper are closer to real-world applications.

**Weaknesses:**

1. The conservative data collection part is actually very similar to the medium data in previous benchmarks, e.g. D4RL, where a fixed policy collects the data with medium-level performance. The paper instead proposes to run the mean of the fixed Gaussian policy 80% of the time and the stochastic Gaussian policy otherwise. I'm not sure if such a change can lead to more practical applications and make a huge difference in the performances of offline RL algorithms.

2. While I generally agree that we need to be more conservative in data collection, I don't think that means we should only collect data with a fixed mostly deterministic policy. Such a setting proposed in the paper could make the data coverage very narrow, thus benefitting BC and model-free methods. I think it would be worth studying the dataset that consists of a mixture of policies or random policies that don't violate certain safety criteria. Those settings should still be practical in real-world tasks and could result in very different outcomes.

**Additional Feedback:**

See the weakness section.

**Correctness:**

The evaluation methods and experimental design seem correct. The claim that the real-world datasets should be collected with a deterministic working policy seems a bit off (see the section above).

**Documentation:**

Yes

**Relation To Prior Work:**

Yes

**Summary And Contributions:**

This paper presents a new offline RL benchmark that considers using a conservative working policy (i.e. deterministic) for data collection in various tasks that simulate real-world applications and also tests the performance of offline evaluation methods. The results show that current offline RL methods don't work well in the benchmark compared to BC, suggesting more improvement is needed.

---

> ### Author Response · Authors · 2022-08-10
> **Response to reviewer uqJ5**
>
> Q1: "The conservative data collection part is actually very similar to the medium data in previous benchmarks, e.g. D4RL, where a fixed policy collects the data with medium-level performance. The paper instead proposes to run the mean of the fixed Gaussian policy 80% of the time and the stochastic Gaussian policy otherwise. I'm not sure if such a change can lead to more practical applications and make a huge difference in the performances of offline RL algorithms."
>
> A1: Although D4RL has datasets collected by running a fixed policy, the policy is stochastic with exploration behaviors. However, we believe the exploration behaviors, if not totally inhibited, cannot happen frequently in real-world systems. The benchmark thus simulates this setting. Section 6.3 Table 2 just compares the achievements of several algorithms on D4RL medium datasets and NeoRL datasets. The results from D4RL would let people believe that the algorithms can achieve significant improvements, which can actually have a much worse performance for the data in the NeoRL setting.
>
>
>
> Q2: "While I generally agree that we need to be more conservative in data collection, I don't think that means we should only collect data with a fixed mostly deterministic policy. Such a setting proposed in the paper could make the data coverage very narrow, thus benefitting BC and model-free methods. I think it would be worth studying the dataset that consists of a mixture of policies or random policies that don't violate certain safety criteria. Those settings should still be practical in real-world tasks and could result in very different outcomes."
>
> A2: We want to clarify that it is not like we unreasonably favor such narrow data distributions. The setting of the narrow data distributions comes from what we have learned in several real-world tasks. We've rarely seen random policies can be used to collect data online. Even if some random actions may not violate certain safety criteria, they always cause a higher cost. Therefore, we eagerly present the challenges in the benchmark and wish to promote real-world applications of RL. Besides, we use 3 policies with similar performance rather than one fixed policy to collect each dataset (please see Line 164-176 for data-collection policies and how we collect data).

---

### Official Review · Reviewer_t3v9 · 2022-07-27
**paper review**

**Rating:** 8
**Confidence:** 3
**Correctness:** The claims in this paper seem correct…
**Clarity:** This paper is well written and easy t…

**Strengths:**

The NeoRL benchmark contains sufficient datasets since it involves 7 domains with 52 tasks in total. Besides, the authors build benchmarks for offline and online policy selection. Moreover, they conducted sufficient experiments to verify the performance of existing offline RL algorithms. And they pointed out some interesting results which can give the right direction for developing offline RL policies in real-world tasks.

**Weaknesses:**

For the NeoRL benchmark, the authors claim that limited samples (1e4) if enough for policy training. However, more training data is helpful to train a more robust policy, especially when we take the security issues of RL policy into account (e.g., adversary attacks).

For the offline validation process, the proposed validation with offline data, before a policy is deployed on a real-world task, indeed improves the efficiency of policy selection. But from Table 1, I noticed that the average rank in online performance is different from the rank in offline performance (e.g., CQL). I am wondering can the offline validation reflect the correct performance rank of a policy in the real-world task? And how to prevent selecting a sub-optimal policy?


**Additional Feedback:**

Typos:
-	Line 87: redundant period

**Documentation:**

The documentation is clear.

**Ethics:**

No ethical concerns.

**Relation To Prior Work:**

This paper well discussed previous works about RL benchmarks and Offline RL algorithms in section 3.

**Summary And Contributions:**

This paper proposes a Near real-world offline RL benchmark, named NeoRL. The NeoRL dataset is collected on different real-world tasks with a more conservative strategy compared with the previous offline RL benchmark. They evaluate some SOTA offline RL algorithms on NeoRL and reveal that current offline RL algorithms are less effective in real-world tasks. The authors propose a new workflow for offline RL policies: offline training, offline validation (new), and online deployment.

---

> ### Author Response · Authors · 2022-08-10
> **Response to reviewer t3v9**
>
> Q1: "the authors claim that limited samples (1e4) if enough for policy training. However, more training data is helpful to train a more robust policy, especially when we take the security issues of RL policy into account (e.g., adversary attacks)."
>
> A1: We would like to clarify that we did not claim the limited samples are enough for policy training. This benchmark limits the sample size (100 trajectories) in order to simulate some real-world situations we encountered where the data is not unlimited, e.g., an urban power network produces about 50 trajectories per year. This setting is a challenge for offline RL algorithms. Once this challenge is well solved, we will have a better solution to deal with a huge range of real-world tasks with limited samples. This is the purpose of the benchmark.
>
>
>
> Q2: "But from Table 1, I noticed that the average rank in online performance is different from the rank in offline performance (e.g., CQL). I am wondering can the offline validation reflect the correct performance rank of a policy in the real-world task? "
>
> A2: Yes, the ranks of online and offline performance are different. This is the state of the current offline validation methods. However, obtaining the online performance of an arbitrary policy can be costly and dangerous, thus better offline validation methods are highly desired. This is also a challenge we want to emphasize in the benchmark. Your question on "how to prevent selecting a sub-optimal policy?" is exactly what this benchmark heads to.
>
>
>
> Q3: "Typos: - Line 87: redundant period"
>
> A3: Thanks for the feedback! We have revised it and updated the PDF.

---

### Author Response · Authors · 2022-08-10
**The general response to all reviewers**

Thanks for your invaluable feedback and insightful suggestions. We appreciate the encouraging comments that all the reviewers think our work is well-written and clear. In the following, we will answer the questions separately. If you have any remaining questions, please inform us!

---

### Meta-Review · Area_Chair_ibDs · 2022-09-02

**Recommendation:** Accept
**Confidence:** 4

**Metareview:**

The paper introduces new datasets and benchmarks for offline settings in RL. The paper's contributions are notable for (1) using and highlighting the importance of learning from narrow data distributions, (2) treating policy evaluation and validation as a key challenge rather than an afterthought, and (3) providing extensive results of existing methods. I believe this paper is a significant contribution to the field and hope that future researchers will use it in addition to the existing offline RL benchmarks.

---

### Decision · Program_Chairs · 2022-09-16

Accept